# Rapid and Convenient Single-Chain Variable Fragment-Employed Electrochemical C-Reactive Protein Detection System

**DOI:** 10.3390/ijms25052859

**Published:** 2024-03-01

**Authors:** Daimei Miura, Saki Motohashi, Ayaka Goto, Hayato Kimura, Wakako Tsugawa, Koji Sode, Kazunori Ikebukuro, Ryutaro Asano

**Affiliations:** 1Department of Biotechnology and Life Science, Graduate School of Engineering, Tokyo University of Agriculture and Technology, 2-24-16 Naka-cho, Koganei 184-8588, Japan; daimei@go.tuat.ac.jp (D.M.); tsugawa@cc.tuat.ac.jp (W.T.); ikebu@cc.tuat.ac.jp (K.I.); 2Joint Department of Biomedical Engineering, University of North Carolina at Chapel Hill and North Carolina State University, Chapel Hill, NC 27599, USA; ksode@email.unc.edu; 3Institute of Global Innovation Research, Tokyo University of Agriculture and Technology, 3-8-1 Harumi-cho, Fuchu 183-8509, Japan

**Keywords:** IgG-free immunosensors, single-chain variable fragment, antibody–enzyme complex, C-reactive protein, point-of-care testing

## Abstract

Although IgG-free immunosensors are in high demand owing to ethical concerns, the development of convenient immunosensors that alternatively integrate recombinantly produced antibody fragments, such as single-chain variable fragments (scFvs), remains challenging. The low affinity of antibody fragments, unlike IgG, caused by monovalent binding to targets often leads to decreased sensitivity. We improved the affinity owing to the bivalent effect by fabricating a bivalent antibody–enzyme complex (AEC) composed of two scFvs and a single glucose dehydrogenase, and developed a rapid and convenient scFv-employed electrochemical detection system for the C-reactive protein (CRP), which is a homopentameric protein biomarker of systemic inflammation. The development of a point-of-care testing (POCT) system is highly desirable; however, no scFv-based CRP-POCT immunosensors have been developed. As expected, the bivalent AEC showed higher affinity than the single scFv and contributed to the high sensitivity of CRP detection. The electrochemical CRP detection using scFv-immobilized magnetic beads and the bivalent AEC as capture and detection antibodies, respectively, was achieved in 20 min without washing steps in human serum and the linear range was 1–10 nM with the limit of detection of 2.9 nM, which has potential to meet the criteria required for POCT application in rapidity, convenience, and hand-held detection devices without employing IgGs.

## 1. Introduction

Antibodies are promising recognition molecules in biosensing due to their high affinity and specificity for targets, specifically in the construction of immunosensors. Immunoglobulin G (IgG) is an antibody type widely used in immunosensors because it has two binding arms, which leads to high sensitivity owing to the bivalent effect. Monoclonal antibodies (mAbs) are often used in immunosensors because of their reproducibility, but almost all mAbs are produced through hybridoma technology, which involves the killing of laboratory animals such as rabbits and mice [1]. Since the European Union introduced the animal testing ban in cosmetic production in 2013, there has been an acceleration with respect to ethical concerns in the production of hybridoma cells, and the EU’s working group finally announced that the generation of antibodies should be shifted from hybridoma cells to recombinant production processes [2]. Therefore, there is an unmet need for the development of IgG-free immunosensors.

Antibody–enzyme complexes (AECs) are promising recognition elements in immunosensors. To date, antibody fragments, such as the single-chain variable fragment (scFv) [3], fragment antibody-binding (Fab) [4], and the variable domain of the heavy chain of the heavy-chain antibody (VHH) [5], have been used to prepare AECs. These antibody fragments can be produced recombinantly using bacterial expression systems; thus, they can be easily modified by means of protein engineering to fit designed immunosensors. However, because these antibody fragments sometimes show a lower affinity than IgGs, some improvement is required to enhance their affinity. Previously, we have reported bivalent AECs composed of a scFv and glucose dehydrogenase (GDH) using the SpyCatcher (SC)/SpyTag (ST) system for affinity enhancement [6], and established an electrochemical detection system by incorporating IgG-immobilized magnetic beads [7,8]. Although highly sensitive detection was achieved in both cases due to the enhanced affinity of the bivalent AEC, IgG was still used to capture the target on the magnetic beads.

When considering scFv immobilization onto a surface instead of IgGs for the development of IgG-free immunosensors, certain studies have focused on the engineering of scFvs because the direct adsorption of antibodies to solid surfaces results in denaturation [9]; the introduction of specific amino acids such as cysteine or histidine for immobilization to the gold surface [10,11], and the fusion of binding peptides such as polystyrene-binding peptides or gold-binding peptides [12,13] have been proposed. The SC/ST reaction forms a covalent bond between SC and ST with high efficiency (Figure 1a), so it is also expected to be a promising tool for the immobilization of scFvs onto certain types of carriers with a maintained binding ability and has already been applied to immobilize VHH on magnetic beads [14]. In addition, the density of antibody fragments immobilized on a surface can be higher than that of the IgG immobilized due to its smaller size [15], which contributes to the high sensitivity [16]. However, this is limited to cases of detection approaches requiring washing procedures such as an enzyme-linked immunosorbent assay (ELISA).

Point-of-care testing (POCT) is a concept for conducting measurements outside of laboratories without specialized technicians. As for immunosensors, bound/free separation such as washing in an ELISA is often required to obtain a target-specific signal, but it takes time and must be performed by well-trained technicians to guarantee the results, which is not suitable for POCT applications. Therefore, detection procedures that do not require a wash step, are convenient in manipulation, and are suitable for use with handheld measurement devices required for POCT immunosensors. Currently, the development of POCT devices to detect protein biomarkers is an intense area of research [17]. Among them, the C-reactive protein (CRP) in serum is a known biomarker of systemic inflammation and has been studied as a predictor of cardiovascular diseases [18,19], and there has been a growing need for CRP-POCT to discern infectious diseases and other inflammatory disorders, which could lead to a reduction in the unnecessary use of antibiotics for treating respiratory tract infections [20]. Serum CRP levels are currently measured using an ELISA, nephelometric immunoassay, or turbidometric immunoassay [21], but they are not applicable to POCT.

In this study, we developed a rapid and convenient electrochemical sandwich immunosensor using a single anti-CRP scFv clone as the capture and detection antibody (Figure 1b). Because CRP is a multimeric protein, only one recombinant protein of the same scFv clone targeting CRP is required for detection without the need for IgGs [22]. In addition, magnetic beads were used, not only to enhance the sensitivity through the immobilization of scFvs via the SC/ST system, but also to eliminate washing procedures, thus simplifying the detection system, which has potential to be applied to CRP-POCT.

## 2. Results

### 2.1. Preparation of Anti-CRP scFv and Anti-CRP scFv-SpyTag

To prepare the anti-CRP scFv and anti-CRP scFv-STs, *Escherichia coli* cells were individually transformed with the corresponding expression vectors. Sodium dodecyl sulfate-polyacrylamide gel electrophoresis (SDS-PAGE) and Western blotting revealed that both the scFv and scFv-ST were expressed in the culture supernatant and intracellular soluble fraction (Appendix A). As the cytoplasm was reduced, leading to misfolding of the scFvs, both the fragments were prepared from the culture supernatant. After Ni-NTA affinity chromatography and gel filtration chromatography, anti-CRP scFv and anti-CRP scFv-ST monomers were successfully fractionated with productivities of 0.31 and 0.03 mg/L culture, respectively (Figure 2a).

### 2.2. Fabrication of AECs

To fabricate the mono- and bivalent AECs, an anti-CRP scFv-ST was stoichiometrically mixed with GDH-SC or SC-GDH-SC and incubated at 4 °C for 18 h. The SDS-PAGE results showed clear bands corresponding to the mono- and bivalent AECs after mixing each component (Figure 2b), suggesting that both AECs were successfully formed. The results showed that the SC/ST system enabled the convenient preparation of AECs with high efficiency through simple mixing and incubation at low temperatures.

### 2.3. Affinity Analysis of the Antibodies by Surface Plasmon Resonance

Antibody affinity was analyzed using surface plasmon resonance (SPR). All the response curves are shown in Figure 3, and the kinetic parameters are summarized in Table 1. The original anti-CRP scFv showed a high affinity for immobilized CRP (dissociation constant, *K*_D_ = 4.8 nM). The fusion of ST slightly decreased the affinity (*K*_D_ = 5.0 nM), and a further decrease in affinity was observed after complex formation with GDH (*K*_D_ = 28.7 nM). In contrast, affinity recovery was observed in the bivalent AEC (*K*_D_ = 6.2 nM). These results indicate that the AECs were successfully fabricated without any functional loss in binding and can contribute to the highly sensitive detection of CRP owing to its enhanced affinity.

### 2.4. Glucose Dehydrogenase Activity Assay of the AECs

The GDH activity of the AECs was evaluated by calculating their specific activity (U/nmol). Here, one unit of specific activity was defined as the amount of enzyme that catalyzes the 1 µmol dichlorophenolindophenol (DCIP) reduction in one min at 25 °C [23]. As shown in Figure 4, an increase in the glucose concentration-dependent enzymatic activity was observed. The kinetic parameters were calculated from each curve and are summarized in Table 2. Neither parameter (*K*_M_ nor *V*_max_) changed drastically after AEC formation, indicating the preparation of the bivalent AEC without a major loss of enzymatic activity.

### 2.5. CRP Detection Using Bivalent AEC and Use of a Single scFv Clone as a Capture and Detection Antibody

To verify whether the bivalent effect contributes to the high sensitivity of CRP detection, we first performed a sandwich ELISA using IgG (clone C2) as a capture antibody and detected it based on the enzymatic activity of the AECs. As expected, a higher signal was observed when bivalent AEC was used as the detection antibody (Figure 5a). This indicates that the bivalency of the detection antibody positively affected multimeric protein detection.

Because CRP is a homopentameric protein, CRP may be recognized by a single scFv clone as a capture and detection antibody in a sandwich ELISA [22]. Therefore, we immobilized anti-CRP scFv onto a 96-well plate and detected CRP using AECs with the same anti-CRP scFv clone. As shown in Figure 5b, the fluorescence intensity increased with CRP concentration when the bivalent AEC was used as the detection antibody. No signal increase was observed when the monovalent AEC was used as the detection antibody. These results clearly show that the bivalent AEC is a potential detection element for highly sensitive immunosensors when a single antibody scFv clone is used as the capture and detection antibody without employing IgGs.

### 2.6. Electrochemical CRP Detection Using Magnetic Beads and the Bivalent AEC

First, we performed the electrochemical CRP detection by following our previous protocol, but the sensitivity was not sufficient for accurate CRP quantification (Appendix A). This was due to the high background signal caused by the non-specific binding of the bivalent AEC. Once the beads were washed after the incubation with CRP and the bivalent AEC, the background signal was significantly decreased (Appendix A).

Therefore, we investigated the incubation conditions to understand how to prevent the non-specific binding of the AEC to the beads. Here, we compared the signal-to-background ratio when the beads were incubated with the AEC and CRP in the presence or absence of 5% skim milk. As shown in Appendix A, the use of 5% skim milk for the incubation decreased the background signal without the CRP while maintaining the signal with CRP compared to the detection in the absence of skim milk. In addition, it was confirmed that the non-specific binding of the bivalent AEC to the beads was not removed by washing three times when incubated without skim milk (Appendix A). Then, we investigated the concentration of the bivalent AEC to increase the response current. As shown in Appendix A, the signal in the presence of the CRP was increased independent of the AEC concentration. On the other hand, the background signal was also increased, and the signal-to-background ratio was not improved. Therefore, we used 200 nM as the concentration of AEC for CRP detection.

Under these conditions, we performed electrochemical CRP detection. The chronoamperogram is shown in Figure 6a. The response current change was dependent on the CRP concentration, and reached a plateau in the high concentration range (Appendix A). The linear range was 1.0–25 nM (1.15 × 10^2^–2.87 × 10^3^ µg/L) and the limit of detection (LOD; 3σ) was 2.4 nM (Figure 6b). Then, we performed the electrochemical CRP detection in human serum to evaluate the feasibility of detection in biological samples. As shown in Appendix A, the CRP concentration-dependent current was increased and also reached a plateau in the high CRP concentration range. The linear range was decreased to 1.0–10 nM compared to the detection in the buffer and the LOD was 2.9 nM, suggesting that the bivalent AEC specifically recognized and detected CRP in the presence of the other biological contaminants within the sample. Taken together, these results suggest that our detection system using scFv-immobilized magnetic beads has potential to be applied to the rapid and convenient detection of CRP-POCT without employing IgGs.

## 3. Discussion

Immunosensors are promising sensing formats, and monoclonal IgG is a widely used antibody as a capture and/or detection antibody, but the production of monoclonal IgG inherently involves the sacrifices of laboratory animals. Therefore, the development of IgG-free immunosensors has become a requirement. To date, various antibody formats such as Fab, scFv, and VHH have been proposed, but their affinities are sometimes lower than that of IgG because these antibodies are monovalent. Our recent research showed that the affinity of these antibodies could be enhanced in combination with a bivalent AEC by focusing on the connection of two scFvs at both ends of GDH via the SC/ST system for immunosensor development [6,7,8], but these immunosensors still used IgG as a capture antibody for high sensitivity. Here, we prepared a bivalent AEC using anti-CRP scFvs and GDH for CRP detection for the development of an IgG-free immunosensor. As shown in Table 1, where the affinity of the monovalent AEC was slightly decreased, that of the bivalent AEC was enhanced, as expected. The affinity decrease in the monovalent AEC could be caused by several factors: (i) interference in the binding by the accompanying GDH, (ii) the remaining SC/ST complex, or (iii) hidden epitopes on the SPR sensor surface. The bivalent AEC showed a decreased *k*_off_, indicating that the bivalent effect contributed to affinity improvement. In addition, the *k*_on_ of the bivalent AEC was also improved compared with that of the monovalent AEC. This may reflect the molecular configuration of the homopentameric CRP protein, increasing the chance of scFv binding to the CRP.

As for the enzymatic activity, the kinetic parameters were not significantly changed after AEC formation (Figure 4 and Table 2). From the crystal structure of GDH from *Aspergillus flavus* (PDB ID: 4YNU), it was found that both terminals were far from the active site of GDH, which allowed for the fusion of SC to both terminals without any impact on the catalytic reaction of GDH. In addition, the SC/ST reaction can proceed with high reactivity even at 4 °C [24]. These factors contributed to the successful preparation of the bivalent AEC with retained catalytic activity comparable to that of the original GDH [25]. The normal blood glucose concentration range is 4.9 to 6.9 mM and that of diabetic patients increases up to 40 mM [26]. This range is lower than the *K*_M_ value of the bivalent AEC (Table 2), suggesting that the response current in the electrochemical detection is influenced by the individual glucose level, which leads to a quantification error unless calibrations with known CRP concentration are performed with every measurement, which is not suitable for POCT application. Therefore, the addition of 200 mM of glucose was required in order to obtain a stable response current from the bivalent AEC.

We confirmed that the bivalent effect in the detection antibody positively affects highly sensitive detection [6]. In particular, it is strongly exerted when the target is a multimeric protein such as hemoglobin [8], and it was also confirmed that the bivalent AEC targeting CRP showed a high sensitivity compared to the monovalent AEC (Figure 5a). As in this case, different antibody clones between capture and detection antibodies were required to perform a sandwich ELISA, which led to an increase in the cost and often the employment of IgGs. For the development of an IgG-free immunosensor, we immobilized anti-CRP scFvs onto a 96-well plate and aimed to achieve CRP quantification using the bivalent AEC for detection. As shown in Figure 5b, CRP was successfully detected without employing IgGs. However, the sensitivity of detection using the scFv as the capture antibody, which can be defined as the slope of the calibration curve [27], was 30 nM^−1^, which was much lower than that when using IgG (106 nM^−1^). This could be caused by the difference in the CRP epitope between the scFv and IgG. Thus, the use of different scFv clones that do not compete with each other as capture and detection antibodies may contribute to highly sensitive detection.

Amperometry-based electrochemical detection often results in rapid and convenient detection. For example, such a sensor could use the same sensor configuration as that used for self-monitoring blood glucose meters, which are the most widely used amperometric detection systems in the world. We demonstrated the electrochemical detection of CRP in a buffer and a human serum, with a linear range between 1.0 and 25 nM (1.15 × 10^2^–2.87 × 10^3^ µg/L) in the buffer and 1.0 and 10 nM (1.15 × 10^2^–1.15 × 10^3^ µg/L) in the human serum without washing. Considering the clinically relevant range of CRP in serum (95–1.9 × 10^3^ nM; 10 × 10^3^–200 × 10^3^ µg/L) for the detection of bacterial infections [28,29] our linear range were within this range after appropriate dilution, but not fully covered because the response current reached a plateau at a higher CRP concentration. This was caused by the limited number of scFvs immobilized on the magnetic beads, which relied on the number of immobilized SCs through the chemical coupling, and on the reaction efficiency between the scFv-ST and the immobilized SC. To extend the linear range for further applications, the immobilization method of SC to the magnetic beads should be investigated. We are now investigating another immobilization method to fix the orientation of SC against magnetic beads for more highly sensitive CRP detection.

Indeed, the washing procedure contributed to the improvement of the signal-to-background ratio (Appendix A), but it took an additional operation time (~10 min), and is not applicable to POCT. The response current depends on the amount of oxidized electron mediators reacted on the electrode surface. Therefore, the electron mediators oxidized by the AEC bound to the target can be reduced more efficiently on the electrode surface than those oxidized by the AEC unbound to the target, which are far from the electrode surface. This is the reason why the washing step was omitted in our detection system, allowing the application to POCT. Table 3 summarizes the characteristics of CRP immunosensors. As CRP is an inflammatory marker, various immunosensors have been proposed. Some IgG-based immunosensors are superior to antibody fragment-based immunosensors in terms of their sensitivity, and lateral flow immunoassay platforms are useful for POCT application; however, immunosensors that do not use IgGs as recognition molecules are limited. Among the IgG-free immunosensors, in terms of their rapidity, convenience, and hand-held detection device, our immunosensor has potential to meet the requirements for POCT application.

## 4. Materials and Methods

### 4.1. Construction of Expression Vectors

A gene of anti-CRP scFv from a previous study by Choi et al. [22] was synthesized after the codon was optimized to express in *E. coli*. To produce the AEC, ST was genetically fused to the C-terminus of an scFv. Each gene was inserted into the T7 promoter-based vector. Expression vectors for SC and SC-fused GDH have been previously constructed [6]. All protein sequences are summarized in Appendix A.

### 4.2. Recombinant Production of Proteins

All the recombinant proteins were prepared using *E. coli* BL21(DE3) as a host. Anti-CRP scFv and scFv-ST antibodies were prepared from the culture supernatant, as previously described. Briefly, the transformants were cultured overnight at 28 °C in Luria-Bertani (LB) broth supplemented with 100 µg/mL of ampicillin, and subsequently cultured in an autoinduction LB medium containing 0.05% glucose, 0.5% glycerol, 0.2% β-lactose, 25 mM of (NH_4_)_2_SO_4_, 50 mM of Na_2_HPO_4_, 50 mM of KH_2_PO_4_, 1 mM of MgSO_4_, and 100 µg/mL of ampicillin at 20 °C and 170 rpm for 40 h. Then, the culture supernatant was treated with 60 wt% ammonium sulfate for salting out, and the protein precipitation was dissolved in 20 mM of potassium phosphate (pH 7.4), followed by dialysis overnight at 4 °C against the same buffer to remove ammonium sulfate. The purification was performed using Ni Sepharose 6 Fast Flow (Cytiva, Tokyo, Japan) with 20 mM of potassium phosphate (pH 7.4) containing 500 mM of NaCl and Superdex 200 Increase (Cytiva) with 20 mM of potassium phosphate (pH 6.5) containing 200 mM of NaCl.

Each protein, SC, GDH-SC, and SC-GDH-SC was prepared from the intracellular soluble fraction of *E. coli* as previously described [6]. Briefly, the transformants were cultured in the autoinduction LB medium containing 50 µg/mL of kanamycin at 37 °C and 170 rpm for 24 h. Cells were disrupted using a French pressure (Ohtake Works, Tokyo, Japan) and the intracellular soluble fraction was collected by a centrifuge at 13,000× *g* for 20 min at 4 °C. Then, the sample was purified using ÄCTA pure™ (Cytiva) and HisTrap HP (Cytiva). The purified fraction was dialyzed against 20 mM of a potassium phosphate buffer (pH 6.5).

### 4.3. Fabrication of AECs

The monovalent AEC was prepared by mixing of 5 µM of anti-CRP scFv-ST with 5 µM of GDH-SC and incubating at 4 °C overnight. The bivalent AEC was prepared by mixing of 10 µM of scFv-ST and 5 µM of SC-GDH-SC and incubating at 4 °C overnight. To evaluate AEC formation, the reaction was terminated by the addition of a buffer containing 125 mM of Tris-HCl (pH 6.5), 20% glycerol, 4% sodium dodecyl sulfate, 3.1% dithiothreitol, and 0.01% bromophenol blue, followed by boiling at 95 °C for 10 min. Then, a SDS-PAGE under reducing conditions was performed, and the gel was stained with Coomassie brilliant blue.

### 4.4. Surface Plasmon Resonance (SPR) Analysis

The affinity between the antibodies and the CRP was analyzed using an SPR analysis (Biacore™ T200, Cytiva). The recombinant human CRP was purchased from Oriental Yeast Co., Ltd. (Tokyo, Japan). First, the amine-reactive moieties in the CRP solution were removed by filtration using an Amicon Ultra-0.5 Centrifugal Filter 3 K (Merck Millipore, Burlington, MA, USA) with 20 mM of a phosphate buffer supplemented with 200 mM of NaCl (pH 6.5). Subsequently, the CRP was diluted to 50 µg/mL using 10 mM of an acetate buffer (pH 4.5) and immobilized onto a Series S Sensor Chip CM5 (Cytiva) using a conventional amine-coupling method of up to 1074 resonance units (RU). A series of concentrations of antibodies (0.50, 1.0, 5.0, 10, 25, 50, and 75 nM) were injected at a flow rate of 30 µL/min, and the association and dissociation times were 120 s. The antibodies bound to the CRP were washed out with 5 mM of NaOH at a flow rate of 10 µL/min for 30 s. The kinetics were calculated by a 1:1 Langmuir global fitting model.

### 4.5. Dehydrogenase Enzyme Activity Assay

The dehydrogenase activities of the GDH-SC, SC-GDH-SC, monovalent and bivalent AECs were evaluated using phenazine methosulfate (PMS) and DCIP as electron mediators. Each 20 µL of enzyme sample solution was mixed with 160 µL of a reaction solution containing 20 mM of potassium phosphate (pH 6.5), 6 mM of PMS, and 0.6 mM of DCIP. Then, 20 µL of glucose at different concentrations was added to initiate the enzymatic reaction, and the absorbance at 600 nm was monitored for 30 s to calculate the kinetic parameters.

### 4.6. Sandwich Enzyme-Linked Immunosorbent Assay (ELISA)

CRP was optically detected via a sandwich ELISA using the scFv as the immobilized antibody and monovalent or bivalent AECs for detection. Anti-CRP scFv antibodies were diluted in a carbonate–bicarbonate buffer (Thermo Fisher Scientific, Waltham, MA, USA) and 500 nM of anti-CRP scFv was immobilized onto a 96-well black Maxisorp™ plate (Thermo Fisher Scientific). Instead of scFvs, anti-CRP IgG was used (clone C2; HyTest Ltd., Turku, Finland). After washing three times, the plate was blocked with a Tris-buffered saline (TBS) buffer containing a 1% (*w*/*v*) bovine serum albumin at room temperature for 1 h. After washing three times, 0–5 nM of CRP was added, and the plate incubated at room temperature for 1 h, then was washed three times. Finally, 175 nM of monovalent or bivalent AEC was added and the plate was incubated at room temperature for 1 h. Finally, after washing six times, a reaction solution composed of 20 mM of a potassium phosphate buffer (pH 6.5), 1 mM of 1-methoxy PMS (mPMS), 500 µM of resazurin, and 200 mM of glucose was added and the plate was incubated at room temperature for 1 min; the fluorescence of resorufin was measured using a plate reader (Varioskan^®^ Flash; Thermo Fisher Scientific; λ_ex_ = 570 nm, λ_em_ = 590 nm).

### 4.7. Electrochemical CRP Detection Using Magnetic Beads

Electrochemical detection using magnetic beads was performed according to our previous reports [7,8] with some modifications. First, SC was chemically immobilized onto NHS-activated magnetic beads (Thermo Fisher Scientific), following the manufacturer’s instructions. Then, 30 µg of SC-immobilized magnetic beads was incubated with 300 nM of anti-CRP scFv-ST at 4 °C for 18 h, and the beads were washed with TBS (pH 8.0) three times to remove unreacted antibodies by magnetic separation. Then, the beads were blocked by incubation with a phosphate-buffered saline containing 5% (*w*/*v*) skim milk under gentle shaking at 25 °C for 1 h, and washed with a TBS containing 0.05% (*v*/*v*) (pH 8.0) three times by magnetic separation.

Next, 30 µg of the scFv-immobilized magnetic beads was mixed with 15 µL of 200 nM of the bivalent AEC diluted with 100 mM of the phosphate buffer (pH 6.5) containing 5% (*w*/*v*) skim milk and with 15 µL of various concentrations of CRP under shaking at 1200 rpm at 25 °C for 15 min. Then, 30 µL of 15 mM of mPMS was mixed, and 20 µL of the mixed solution was subsequently loaded onto a screen-printed carbon electrode (DEP-Chip EP-PP, BioDevice Technology Ltd., Ishikawa, Japan), where a magnet was set under the electrode to accumulate the magnetic beads on the working electrode. Chronoamperometry was performed at a potential of +150 mV vs. Ag/AgCl. After 5 min, 5 µL of 500 mM of glucose was spiked and the current change was monitored. The current increase was plotted 40 s after the glucose addition. As for the CRP detection in the human serum, we diluted CRP with the pooled human serum purchased from Cosmo Bio Co., Ltd., and incubated the serum containing the diluted CRP with the bivalent AEC diluted with 100 mM of the phosphate buffer (pH 6.5) containing 5% (*w*/*v*) skim milk and the magnetic beads. Other detection methods were the same as the detection in the buffer.

## 5. Conclusions

In conclusion, we prepared a bivalent AEC composed of anti-CRP scFv-ST and SC-GDH-SC and applied it to electrochemical CRP detection. The bivalent AEC contributed to the highly sensitive detection of the CRP owing to the bivalent effect, and a rapid and convenient electrochemical detection system was established using scFv-immobilized magnetic beads without IgGs, which has potential to be applied to the development of CRP-POCT applications in the near future.

## Figures and Tables

**Figure 1 ijms-25-02859-f001:**
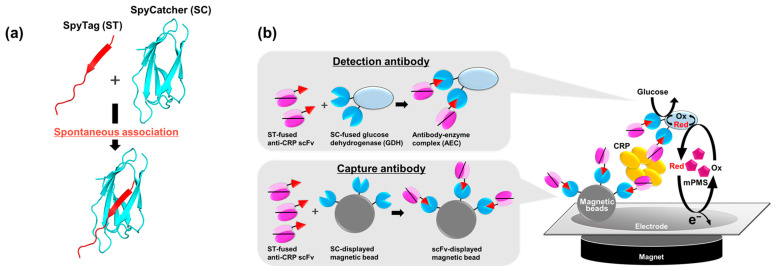
(**a**) An illustration of the association of SpyCatcher (SC) and SpyTag (ST). This was generated from a crystal structure of the SC/ST complex (PDB ID: 4MLI) using the PyMOL Molecular Graphic System, Version 2.5.2, Schrödinger, LLC. (**b**) A schematic illustration of electrochemical C-reactive protein (CRP) detection using anti-CRP single-chain variable fragment (scFv)-immobilized magnetic beads as a capture antibody and an antibody–enzyme complex (AEC) composed of anti-CRP scFvs and glucose dehydrogenase (GDH) as a detection antibody. The magnetic beads presenting anti-CRP scFvs and the AEC were prepared through the SC/ST reaction. mPMS, 1-methoxy PMS; Ox, oxidized form; Red, reduced form.

**Figure 2 ijms-25-02859-f002:**
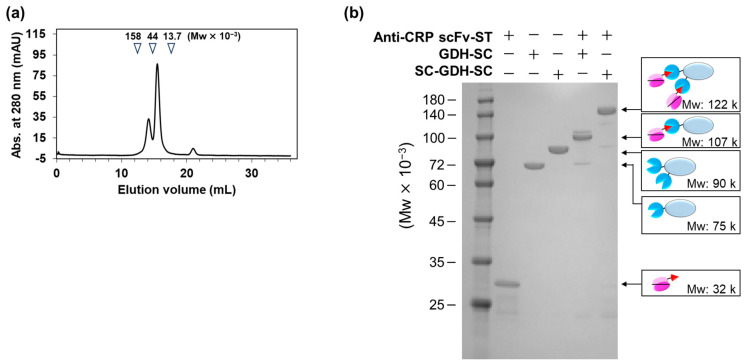
The preparation of anti-CRP scFv and the mono- and bivalent AECs. (**a**) A chromatogram after gel filtration chromatography for the purification of anti-CRP scFv-STs. The elution volume of each marker and its molecular size is shown above. (**b**) SDS-PAGE analysis of each sample for confirmation of AEC formation using the fractionated anti-CRP scFv-ST monomers. CRP, C-reactive protein; scFv, single-chain variable fragment; ST, SpyTag; SDS-PAGE, sodium dodecyl sulfate-polyacrylamide gel electrophoresis; AEC, antibody–enzyme complex.

**Figure 3 ijms-25-02859-f003:**
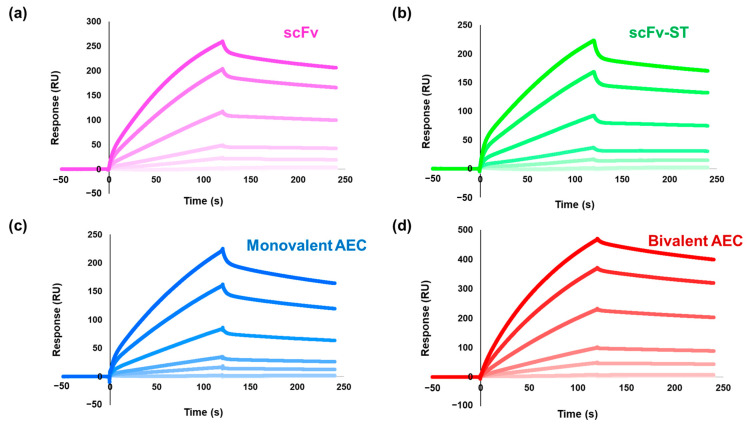
The affinity analysis of the prepared antibodies by surface plasmon resonance. The analyzed sensorgrams of (**a**) anti-CRP scFv, (**b**) anti-CRP scFv-ST, (**c**) the monovalent AEC, and (**d**) the bivalent AEC. The darker the color of each plot, the higher the concentration of each protein used for the analysis. The detailed concentrations used for the analysis are described in Materials and Methods. ScFv, single-chain variable fragment; AEC, antibody–enzyme complex; CRP, C-reactive protein.

**Figure 4 ijms-25-02859-f004:**
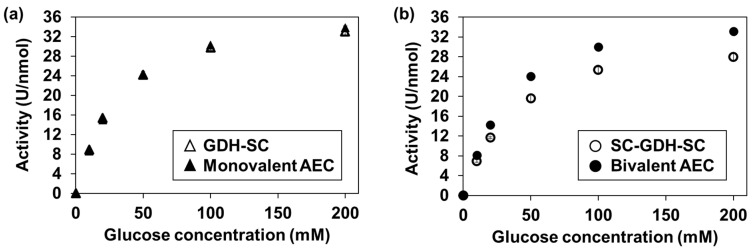
Glucose dehydrogenase activity assay. The amount of enzyme which can catalyze the reduction of 1 µmol of the final electron acceptor (dichlorophenolindophenol; DCIP) in one min at 25 °C was defined as one unit (U). The enzymatic activity using glucose as the substrate was compared before and after the formation of (**a**) the monovalent AEC, and (**b**) the bivalent AEC. The results before and after AEC formation are shown as open and filled symbols, respectively. The data are shown in mean ± S.D. (*n* = 3). AEC, antibody–enzyme complex; GDH, glucose dehydrogenase.

**Figure 5 ijms-25-02859-f005:**
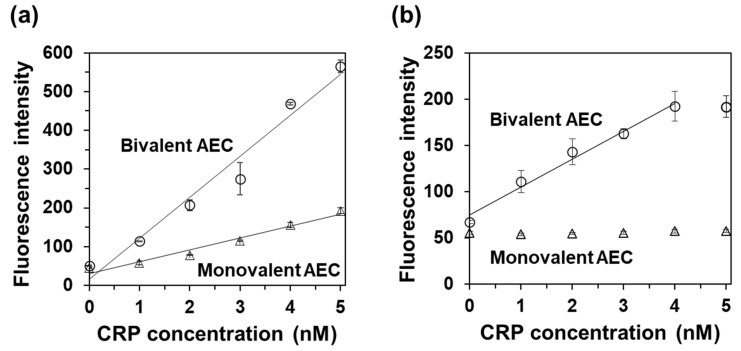
Sandwich ELISA using anti-CRP scFvs as the immobilized and detection antibody. (**a**) The detection of CRP using the monovalent or the bivalent AEC as the detection antibody and anti-CRP IgG (clone: C2) as the capture antibody. (**b**) The detection of CRP using the monovalent or the bivalent AEC for detection and anti-CRP scFv as the capture antibody. The fluorescence signals when using the monovalent or bivalent AEC are represented by triangles and circles, respectively. The measurements at each CRP concentration were performed in triplicate, and the data are shown as the mean ± S.D. (*n* = 3). [(**a**) R^2^ = 0.97 for both the monovalent and bivalent AEC, respectively; (**b**) R^2^ = 0.98 for the bivalent AEC]. CRP, C-reactive protein; AEC, antibody–enzyme complex.

**Figure 6 ijms-25-02859-f006:**
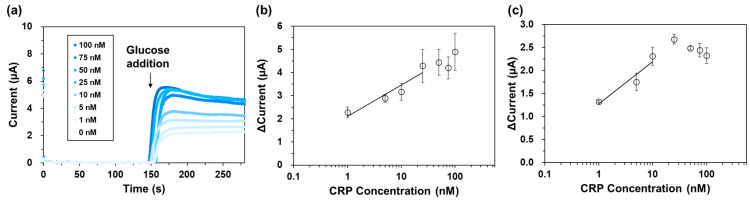
Electrochemical CRP detection. (**a**) Chronoamperograms of each CRP concentration. Calibration curves of the electrochemical CRP detection (**b**) in the buffer and (**c**) in the human serum. Here, ΔCurrent was calculated from the equation, ΔCurrent = I + glu − I-glu. I + glu represents the response current value after glucose injection, and I-glu is before glucose injection. Data are shown as the mean ± S.D. (*n* = 3; R^2^ = 0.97 and 0.98 in the buffer and human serum, respectively). CRP, C-reactive protein.

**Table 1 ijms-25-02859-t001:** The binding parameters of the antibodies to immobilized CRP by surface plasmon resonance. scFv, single-chain variable fragment; ST, SpyTag; AEC, antibody-enzyme complex.

	*k*_on_ (×10^5^, M^−1^s^−1^)	*k*_off_ (×10^−3^, s^−1^)	*K*_D_ (×10^−9^, M)	Chi^2^
scFv	3.5 ± 0.024	1.6 ± 0.0070	4.8 ± 0.039	2.2
scFv-ST	3.4 ± 0.061	1.7 ± 0.016	5.0 ± 0.10	3.9
Monovalent AEC	0.67 ± 0.0060	1.9 ± 0.0099	28.7 ± 0.29	2.4
Bivalent AEC	1.8 ± 0.011	1.1 ± 0.0040	6.2 ± 0.045	6.2

**Table 2 ijms-25-02859-t002:** The kinetic parameters of the enzymes using glucose as the substrate. GDH, glucose dehydrogenase; SC, SpyCatcher; AEC, antibody–enzyme complex.

	*K*_M_ (mM)	*V*_max_ (U nmol^−1^)
GDH-SC	36	33
Monovalent AEC	34	39
SD-GDH-SC	30	38
Bivalent AEC	33	40

**Table 3 ijms-25-02859-t003:** Comparison of IgG-based and antibody fragment-based CRP immunosensors. N.A.: not available.

Used Antibody Format	Detection Technique	Linear Range (µg/L)	LOD (µg/L)	Required Time	POCT Application	Comment	Ref.
IgG	ELISA	0–2 (in serum)	0.032 (in serum)	3 h	Difficult	It is the same configuration of a conventional ELISA.	[30]
IgG	Lateral flow immunoassay	100–5000 (in buffer)	1 (in buffer)	15 min	Possible	Detection was performed by naked eyes.	[31]
IgG	Lateral flow immunoassay with electrochemiluminescence	0.01–1000 (in serum)	4.6 × 10^−3^ (in serum)	15 min	Possible	Electrochemiluminescence image analysis is required on paper-based sensor strip.	[32]
scFv (Cys mutant)	Localized surface plasmon resonance	1–10,000 (in serum)	N.A.	N.A.	Difficult	A large scale and sophisticated instrument is required.	[33]
scFv	Turbidimetry	0–2960 (in buffer)	N.A.	5 min	Difficult	It was detected by a clinical laboratory scale analyzer.	[34]
scFv and Ru^+^-complex labeled scFv	Electrochemiluminescence	0.005–600 (in buffer)	3.0 × 10^−6^ (in buffer)	30 min	Difficult	A large-scale instrument is still required.	[21]
VHH	Electrochemical impedance spectroscopy	250–1000 (in buffer)	210 (in buffer)	N.A.	Difficult	Washing procedures must be required.	[35]
Fluorescence Labeled Fab	Fluorescence polarization	N.A.	207 (in buffer)	10 min	Difficult	A sophisticated optical instrument is required.	[36]
scFv and AEC composed of scFv and GDH	Amperometry	115–2870 (in buffer) 115–1150 (in serum)	276 (in buffer) 333 (in serum)	20 min	Possible	An established glucose sensor-like strip can be used.	This work

## Data Availability

All the data or materials that supported the findings of this study are available by the corresponding author upon request.

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
