# Peer review of "Rapid and Convenient Single-Chain Variable Fragment-Employed Electrochemical C-Reactive Protein Detection System"

_ijms, 2024, doi:10.3390/ijms25052859_

Round 1

Reviewer 1 Report

Comments and Suggestions for Authors

Antibodies are a widely used tool for research, diagnosis and therapy of various diseases. Replacement of antibodies obtained by the hybridoma method with recombinant antibodies and their fragments are of great interest. In this study, an electrochemical sensor for C-reactive protein (CRP) determination using the bivalent antibody-enzyme complex (AEC) composed of anti-CRP scFv and glucose dehydrogenase was developed. The sandwich complex, consisting of capture antibodies (scFv), analyte and detection antibodies (bivalent AEC), was concentrated using magnetic nanoparticles. The article is well structured and clearly presents the results obtained. This manuscript includes studies of the protein interaction affinity of scFv and AEC, as well as the dehydrogenase activity of AEC. However, publication of this manuscript in IJMS is recommended after consideration of the following comments:

1) Line 159. The definition of DCIP should be given in the text where it first appears.

2) Please reconsider the capture to Figure 5a, as the “schematic illustration of the ELISA” is not correct in this case. Perhaps this is the dependence of the fluorescence intensity on the CRP concentration obtained by ELISA using IgG (clone C2) as a capture antibody and AECs as detection antibody.

3) Line 259-260 “However, the sensitivity of detection using scFv as the capture antibody was 30 nM-1, which was much lower than that of detection using IgG (106 nM-1)”. Please explain where the data on the sensitivity of CRP determination came from and how they were obtained (calculated).

4) The authors assessed the linear range of protein determination using three points (Figure 6b). However, although three points is the minimum for a linear fit, most methods require a minimum of five points for a linear fit. In this case, it is recommended to supplement the calibration curve with measurements of CRP concentrations within the specified linear range, since there is a high background signal in the system and the amplitude of the signal fluctuation is low (approximately 200 nA).

5) The manuscript does not contain data on optimizing the concentrations of components for ELISA, as well as electrochemical determination of protein. How were the optimal concentrations of bivalent AEC, Anti-CRP scFv determined?

6) The capture to Figures S1b and S1c does not contain data on CRP concentrations for which chronoamperograms were obtained after the washing stage.

7) Given the presence of data on the positive contribution of the stage of washing magnetic particles to the background signal (its reduction), it is not clear why the authors did not obtain a calibration curve under optimized conditions and limited the study to only presenting the results in supplementary material. Obtaining a calibration curve under optimized conditions would greatly improve this study.

Reviewer 2 Report

Comments and Suggestions for Authors

The article by Daimei Miura et al. presents extensive preparatory work towards developing a CRP detection system. However, despite the authors' optimistic claims, their efforts to demonstrate the system's efficacy for electrochemical CRP detection appear to be unsuccessful.

A significant concern arises from the calibration curve in Figure 6b. The data indicates that all tested concentrations, ranging from 2 to 400 nM, yield indistinguishable values according to the three sigma criterion. This inability to differentiate between concentration levels suggests that the detection limit is not effectively established, undermining the feasibility of the proposed approach for creating practical test systems.

Moreover, the specificity of the system in relation to other substances remains unexplored. This omission raises questions about the potential cross-reactivity of the system, which is a critical factor in determining the accuracy and reliability of any diagnostic tool.

Another notable limitation is the absence of validation using real samples, such as human blood serum. The practical applicability of any diagnostic system, especially for point-of-care testing, hinges on its performance under real-world conditions. The lack of such data severely limits the credibility of the system's utility in clinical settings.

Given these shortcomings, the title contains the "Electrochemical Detection System for C-Reactive Protein Applicable to Point-of-care Testing" appears to overstate the findings. The results, as they currently stand, do not convincingly support the system's applicability for POC testing of CRP.

Reviewer 3 Report

Comments and Suggestions for Authors

After reviewing the manuscript entitled “Rapid and convenient single-chain variable fragments-employed electrochemical detection system for C-reactive protein applicable to point-of-care testing” seems to be an interesting article. However, there are some points to be considered before recommending it for publication:

1. The authors should rewrite the second and third paragraphs of the Introduction (lines 49-78) because they are confusing, and the explanation is unclear.

2. The authors should highlight why the washing steps using this methodology are not necessary, since it is a novelty that saves a lot of time. In addition, they should also comment if they have tried this same procedure carrying out these washing steps.

3. In Section 2.2. Fabrication of the AECs, authors should include how long mixtures anti-CRP scFv-ST with GDH-SC or SC-GDH-SC are incubated.

4. The authors should discuss whether they have studied whether glucose naturally present in biological samples could interfere with the sensor's operation.

5. For future articles, the authors should analyze the analyte content (C-reactive protein in this case) in a certified reference material or a real biological simple.

Comments on the Quality of English Language

Simpler phrases should be used

Reviewer 4 Report

Comments and Suggestions for Authors

The manuscript « Rapid and Convenient Single-chain Variable Fragments-Employed Electrochemical Detection System for C-Reactive Protein Applicable to Point-of-care Testing” by Muira et al., presents an electrochemical biosensor based on antibody fragment for detection of C-reactive protein.

Single chain variable fragment is employed instead of antibody. In addition, 2 fragments were attached to glucose dehydrogenase in order to improve the affinity of binding to the target and to enable electrochemical read-out, respectively.

Abstract should be rewritten to straighten the novelty of the work.

For instance, instead of

“the development of convenient immunosensors that alternatively integrate recombinantly produced antibody fragments, such as single-chain variable fragments (scFvs), remains challenging.” The challenge that was addressed should be named.

Instead of

“detection was achieved in 25 min without washing steps, and the linear range covered the clinically required range. “

The linear ranged should be provided in Abstract.

In Introduction lines 89-xx, washing is not a problem when MPs are used because washing can be performed without the need for electricity source. MELISA, ELISA using magnetic beads, are well established and commercialized approach. So, this is not the novelty of the work.

The main novelty seems to be the employment of 2 antibody fragments instead of one. This was obtained by mixing 2-time concentrated solution of fragments with glucose dehydrogenase. However the detection of CRP by using antibody or this complex with 2 fragments has not be compared.

The calibration curve in Fig. 6 should have more concentrations in the linear range (at least 5 are needed). Also a negative control should be presented.

These controls experiments are needed to justify the utilization of 2 single chain fragments.

Round 2

Reviewer 1 Report

Comments and Suggestions for Authors

The authors adressed all comments, so the manuscript is recommended for publication.

Author Response

We appreciate the reviewer for taking the time to revise our manuscript and giving the oppotunity to receive the feedback.

Reviewer 2 Report

Comments and Suggestions for Authors

The authors have made some updates to the manuscript, yet the revisions do not substantially alter the work's focus or re-evaluate its findings. In its current state, the manuscript might not meet the publication standards of a high-quality journal such as IJMS, primarily due to discrepancies between the presented results and the interpretations provided by the authors.

A critical concern is the apparent contradiction between the results and the claims made in the title, abstract, and discussion sections. For instance, the abstract asserts: "the linear range of 1–100 nM CRP covers the clinically required range even in human serum," a claim reiterated in section 2.6 of the results and discussion. Contrary to these assertions, the data depicted in Figure 6c suggest that the signal does not vary with concentration within the 10–100 nM range. Similarly, in a buffer solution (as shown in Figure 6b), the signal remains constant across a 30-100 nM range, undermining the claimed advantage of the proposed system.

Furthermore, the detection limit presented by the authors falls short of those achieved by existing commercial point-of-care systems and other contemporary research findings. The lack of a comparative analysis table, which is standard for such studies, prevents a clear understanding of how the proposed system stacks up against current benchmarks.

The work's analytical characteristics, including a limited linear range (not exceeding 1-10 nM) and an unremarkable detection limit, do not support the bold claims made in the manuscript. The title, "Detection System for C-Reactive Protein Applicable to Point-of-care Testing," suggests a broader applicability and performance than what the data substantiates.

Therefore, I maintain that this work is not suitable for publication without a revision of its philosophy, a reduction in the extent of its claims, and the inclusion of a more critical discussion of results.

Reviewer 3 Report

Comments and Suggestions for Authors

Thank you very much for making the requested modifications.

Author Response

(The authors gave the same response as above.)

Reviewer 4 Report

Comments and Suggestions for Authors

The manusript has been significantly improved. 

Author Response

(The authors gave the same response as above.)

Round 3

Reviewer 2 Report

Comments and Suggestions for Authors

Thank you for revising your manuscript. However, additional modifications are necessary for publication: - Include the sensor's detection limit and linear range in serum in the abstract. - Move Table S1 from the supplementary to the main text, indicating the detection limit and the matrix type (buffer/serum). - Enhance Table S1 by adding a comparison with the best non-scFvs-based POC methodologies for CRP assays, including the most advanced lateral-flow approaches.
